# Application of Quantum—Markov Open System Models to Human Cognition and Decision

**DOI:** 10.3390/e22090990

**Published:** 2020-09-04

**Authors:** Jerome Busemeyer, Qizi Zhang, S. N. Balakrishnan, Zheng Wang

**Affiliations:** 1Department of Psychological and Brain Sciences, Indiana University, Bloomington, IN 47405, USA; 2Department of Mechanical and Aerospace Engineering, Missiouri University of Science Technology, Rolla, MO 65401, USA; qzwtb@mst.edu; 3Department of Mechanical Engineering, Missiouri University of Science Technology, Rolla, MO 65401, USA; bala@mst.edu; 4Center for Brain and Cognitive, Translational Data Analytics Institute, School of Communication, The Ohio State University, Columbus, OH 43210, USA; wang.1243@osu.edu

**Keywords:** Markov random walk, quantum walk, open system models, evidence accumulation, preference accumulation, choice behavior, decision time, confidence

## Abstract

Markov processes, such as random walk models, have been successfully used by cognitive and neural scientists to model human choice behavior and decision time for over 50 years. Recently, quantum walk models have been introduced as an alternative way to model the dynamics of human choice and confidence across time. Empirical evidence points to the need for both types of processes, and open system models provide a way to incorporate them both into a single process. However, some of the constraints required by open system models present challenges for achieving this goal. The purpose of this article is to address these challenges and formulate open system models that have good potential to make important advancements in cognitive science.

## 1. Introduction

One of the most fundamental topics in the cognitive and neural sciences concerns the dynamic and stochastic processes that humans (as well as other animals) use to make choices and decisions. Consider the following example of what is called a signal detection task: Suppose a radiologist is examining an image of a breast and trying to decide whether or not a cancerous node is present. The process requires accumulating evidence across time (by looking at various parts of the image) until a sufficient amount of evidence has been accumulated to make a decision. However, the decision also depends on the consequences that can occur, which depend on the true state and final choice. For example, missing a cancerous node may allow a cancer to grow into a more dangerous medical problem and falsely deciding that cancer is present produces a great deal of unnecessary stress and additional medical testing. The decision process is fundamentally probabilistic in the sense that if the same radiologist is presented the same image on two different occasions (separated in time with other images in between), she might make different decisions. In addition, the decision on each occasion takes time and the time to make the decision varies across occasions too. Finally after a decision is made, the radiologist could be asked to report how confident she is about her decision. Therefore the basic measures that are collected in a signal detection task are the probability of making each choice and confidence rating, and the distribution of decision times for each choice.

For over 50 years these types of decisions have been successfully modeled by cognitive scientists (see, e.g., [1]) and more recently also by neuroscientists (see, e.g., [2]) using Markov processes, such as random walk (discrete) or diffusion (continuous) processes. The general idea is similar to a Bayesian sequential sampling decision rule [3]. Using the radiologist example, the decision maker starts with some initial state of evidence (e.g., log likelihood) for or against the cancer hypothesis, denoted L(0). During each moment in time *t*, the decision samples evidence, denoted X(t), which increases or decreases the current state, L(t)=L(t−1)+X(t). This evidence continues to be accumulated in the direction of the mean evidence μ=E[X(t)] until its magnitude exceeds a threshold bound θ at which point in time, say *T*, the decision maker stops and decides that cancer present if L(T)>θ or decides cancer is not present if L(T)<−θ. These Markov models of evidence accumulation provide very accurate predictions for empirical distributions of choice and decision times from human decision makers [1] as well as predicting neural activation trajectories from electrophysiological recordings of neurons in primates [2].

Despite the success of Markov processes for modeling these types of signal detection tasks, there is empirical evidence that this class of model may not provide a complete picture of human decision making. Recent research suggests that an alternative way to model evidence accumulation, based on quantum walks, is also needed [4,5,6,7]. (This article is focused on applications of quantum dynamics to human decision making. There are many other applications of quantum theory to decision making that do not involve dynamics, which are not reviewed here. See [8] for a review. Quantum theory has also been applied to strategic games, see [9] for an interesting example, but this work is outside the scope of this article.)

One line of empirical evidence supporting quantum models comes from interference effects of choice on later confidence. In one experiment [5], a signal detection task was used to compare the results from two different conditions: (1) A choice-confidence condition under which participants started observing an image at $t_{0}$ and made a binary decision regarding the image at time $t_{1}$, and then continued to observe the image until they made a confidence rating at time $t_{2}$, and (2) a confidence-alone condition under which participants again started observing an image at $t_{0}$, but simply made a pre-planned button push (to control for responding while not requiring any decision about the image) at time $t_{1}$, and then continued to observe the image until they made a confidence rating at time $t_{2}$. The critical test concerns the distribution of confidence ratings observed at time $t_{2}$ (pooled across choices at time $t_{1}$ for the choice-confidence condition). A Markov random walk model predicts no difference between conditions because it satisfies the Chapman–Kolmogorov equation and the quantum walk predicts differences, because a wave function “collapse” occurs at time $t_{1}$ under the choice-confidence condition but not under the confidence-alone condition. The results of the experiment found significant interference effects, contrary to the Markov model and supporting the quantum model. A second follow-up study, again using confidence ratings, found further support for the quantum walk over the Markov random walk [6].

A second line of empirical evidence supporting quantum models comes from temporal oscillations in preference. In one experiment [7], a preference task was used to investigate how preferences evolve across deliberation time. Participants were presented a choice between two different gift coupons for restaurants, which varied according attributes including the quality of the restaurant, the distance to the restaurant, and the monetary amount of the gift card. They rated their strength of preference for one gift over the other across a series of time points. Markov random walk models used in cognitive science to model preferences (see, e.g., [10]) predict that mean preference strength should monotonically increase across time in the direction of the alternative with greater mean utility. In contrast, a quantum walk model predicts that preferences should oscillate while moving in the direction of the alternative with greater mean utility. The results of the experiment found significant oscillation effects, contrary to the Markov model and supporting the quantum model.

In addition to these lines of evidence, quantum dynamics have been used to account for violations of rational decision making [11], as well as several dynamic decision inconsistencies [12].

In sum, properties of both Markov and quantum walk models may be needed to capture the full probabilistic dynamics underlying human choice and decision making. More fundamentally, Markov and quantum models represent two different types of uncertainty in the decision process [13]. Markov models represent an epistemic type of uncertainty in which an outside observer is uncertain about the internal location of evidence of the decision maker. Quantum models represent an ontic type of uncertainty in which there is no preexisting location of evidence before a decision is made. Instead, at each moment, the decision maker is in a superposed state with several different levels of evidence having potential to be realized, so that the decision maker has internal uncertainty about the level of evidence. Open system models are ideally suited for combining these two different types of dynamics into a single unified process [11,14,15,16,17].

Open system models were developed to represent quantum systems that are described by a combination of a target system of interest coupled to a complex and uncontrolled environment [18]. The original system-plus-environment model relies on purely unitary dynamics that generate interactions between the system and environment. The open system dynamics is derived from unitary dynamics by marginalizing (partial tracing) over the environment to focus on the dynamics of the system alone. The dynamics of the resulting open system starts out in a quantum regime in a“coherent” superposition state however, the dynamics produced by the environment produce decoherence and eventually reduces the system to a classical probability mixture that evolves according to a classical Markov system.

Methods for constructing open system models for applications in physics have been very thoroughly developed. But how can this work in cognitive science? The purpose of this article is to provide some guidelines for applying open system models to cognitive science in a compelling and effective manner.

## 2. Results

Before jumping into the general open system model, it may be helpful to first review versions of Markov and quantum walk processes in isolation. Both quantum and Markov processes can be developed using any combination of discrete versus continuous state, and discrete versus continuous time assumptions. For example, a standard random walk is a discrete state and time Markov chain, and the diffusion model is a continuous state and time Markov process, and quantum models for position and momentum are continuous state and time processes, but the “coin” quantum walk (for review, see [19]) is a discrete time and state model. Furthermore, note that as the increments between states and time steps decrease in size, the discrete models converge to the continuous models (see, e.g., Ref. [20], for Markov processes and Ref. [21] for quantum processes). For ease of presentation, we will work with Markov and quantum walks that are discrete state and continuous time. We present the two different classes of models in a parallel manner to illustrate their common and distinctive features.

### 2.1. Comparison of Markov and Quantum Walk Models

Consider a model in which the decision maker represents their beliefs within a N=101 dimensional vector space. The 101 basis vectors (eigenstates), symbolized as $\left\{|0\rangle ,|1\rangle ,\ldots ,|99\rangle ,|100\rangle \right\}$, represent 101 increasing levels of evidence for one hypothesis over another. Using the radiologist example, the basis vector $|0\rangle$ represents 0.00 likelihood that cancer is present (1.0 likelihood that there is no cancer), $|35\rangle$ represents a 0.35 likelihood favoring cancer, $|50\rangle$ represents equal likelihood, $|65\rangle$ represents a 0.65 likelihood favoring the cancer, and $|100\rangle$ represents a 1.0 likelihood favoring cancer (0.00 likelihood of no cancer). This fine grain evidence scale provides a close approximation of the finite state model to a continuous state model. (Cognitive science models, see, e.g., [1], often use a continuous scale of evidence).

Using the evidence basis to form a belief state, each basis vector can be assigned a coordinate value. For a Markov model, each coordinate is a probability, $\phi_{j}$, representing the probability that the system is located at that level of evidence. For a quantum model, each coordinate is a complex amplitude, $\psi_{j}$ representing the potential to observe that level of evidence. The 101 coordinate values form a 101×1 column matrix denoted here as ϕ for a Markov model and ψ for a quantum model.

Suppose ξ is an arbitrary vector in the space. A Markov model uses an $L_{1}$ norm defined by ${∥\xi ∥}^{1}=\sum_{j}{\left|\xi_{j}\right|}^{1}$ to define length, and requires ${∥\phi ∥}^{1}=\sum_{j}{\left|\phi_{j}\right|}^{1}=1$. In other words, the probabilities must sum to one. A quantum model uses an $L_{2}$ norm defined by ${∥\xi ∥}^{2}=\sum_{j}{\left|\xi_{j}\right|}^{2}$ and requires ${∥\psi ∥}^{2}=\sum_{j}{\left|\psi_{j}\right|}^{2}=1$. In other words the squared magnitudes of the amplitudes must sum to one.

A measurement in this space is represented by a projector. With respect to the evidence basis, the projector is simply a diagonal matrix, denoted as $M_{R}$ for response *R*, with zeros everywhere except one’s located at levels of evidence that represent the response. For example, a response to report a confidence rating equal to 0.65 could be represented by a diagonal matrix $M_{65}$ with a one located at the row 66 (corresponding to the basis vector $|65\rangle$) and zeros elsewhere. A response to decide cancer is present could be represented by a diagonal matrix $M_{C}$ with one’s located at rows 51 to 100 and zeros otherwise. For a Markov model, the probability of a response *R* is given by $p\left(R\right)={∥M_{R}\cdot \phi ∥}^{1}$. For a quantum model, the probability of a response *R* is given by $p\left(R\right)={∥M_{R}\cdot \psi ∥}^{2}$.

The state ϕ(t) of a Markov model evolves across time according to the Kolmogorov forward equation $\frac{d}{dt}\phi \left(t\right)=K\cdot \phi \left(t\right)$ (assuming time invariance), where *K* is the generator or intensity matrix. The solution to this differential equation is $\phi \left(t\right)=e^{K\cdot t}\cdot \phi \left(0\right)$, where $T\left(t\right)=e^{K\cdot t}$ is the transition matrix for time *t*. The generator *K* must have positive off diagonal entries and columns that sum to zero in order to produce a transition matrix T(t). The transition matrix T(t) must contain transition probabilities that sum to unity within each column to generate a new state ϕ(t) containing probabilities that sum to unity.

The state ψ(t) of a quantum model evolves across time according to the Schrödinger equation $\frac{d}{dt}\psi \left(t\right)=-i\cdot H\cdot \psi \left(t\right)$ (assuming time invariance), where *H* is the Hamiltonian matrix. The solution to this differential equation is $\psi \left(t\right)=e^{-i\cdot H\cdot t}\cdot \psi \left(0\right)$, where $U\left(t\right)=e^{-i\cdot H\cdot t}$ is a unitary matrix for time *t*. The Hamiltonian matrix *H* must be Hermitian in order to produce a unitary operator U(t). The unitary matrix U(t) must be an orthonormal matrix in order to generate a new state ψ(t) containing amplitudes with squared magnitudes that sum to unity.

For example, according to a Markov model, the probability of deciding cancer is present at time $t_{1}$ and then giving a confidence rating equal to 65 at time $t_{2}$ equals: $p\left(R\left(t_{1}\right)=R_{C},R\left(t_{2}\right)=R_{65}\right)={∥M_{65}\cdot T\left(t_{2}-t_{1}\right)\cdot M_{C}\cdot T\left(t_{1}-t_{0}\right)\cdot \phi \left(0\right)∥}^{1}$ and according to a quantum model, $p\left(R\left(t_{1}\right)=R_{C},R\left(t_{2}\right)=R_{65}\right)={∥M_{65}\cdot U\left(t_{2}-t_{1}\right)\cdot M_{C}\cdot U\left(t_{1}-t_{0}\right)\cdot \psi \left(0\right)∥}^{2}$. Essentially, a Markov model operates with a transition matrix on probabilities and uses the $L_{1}$ norm, but a quantum model operates on amplitudes with a unitary operator and uses the $L_{2}$ norm.

### 2.2. Representation of States by Density Operators

We can reformulate a pure quantum process using states described by a density operator instead of a state vector. A state vector ψ can be turned into a density matrix ρ by forming the projector $\rho =\psi \cdot \psi^{†}$. The quantum evolution of the density matrix is then given by $\frac{d}{dt}\rho \left(t\right)=-i\cdot \left[H\cdot \rho (t)-\rho (t)\cdot H\right]=-i\cdot \left[H,\rho (t)\right]$ with solution $\rho \left(t\right)=U\left(t\right)\cdot \rho \left(0\right)\cdot U{\left(t\right)}^{†}$. The advantage of this state representation is that it provides a more general formulation of the state by encompassing a probability mixture across pure states, $\rho \left(t\right)=\sum_{j}p_{j}\cdot \psi_{j}\cdot \psi_{j}^{†}$. By linearity, this more general density matrix continues to follow the same quantum evolution equation $\frac{d}{dt}\rho \left(t\right)=-i\cdot \left[H,\rho (t)\right]$. The density matrix thus contains two different types of uncertainty: Epistemic uncertainty in which an outside observer is uncertain about the state of the decision maker, and ontic uncertainty in which the decision maker is in a superposition state over evidence.

The diagonal entries of the density matrix contain the probabilities of observing the N=101 evidence levels. The probability of a response *R* is now computed by the trace $p\left(R\right)=Tr\left[M_{R}\cdot \rho \cdot M_{R}^{†}\right]$. For example, $p\left(R\left(t_{1}\right)=R_{C},R\left(t_{2}\right)=R_{65}\right)=Tr\left[M_{65}\cdot U\left(t_{2}-t_{1}\right)\cdot M_{C}\cdot U\left(t_{1}-t_{0}\right)\cdot \rho \left(0\right)\cdot U^{†}\left(t_{1}-t_{0}\right)\cdot M_{C}^{†}\cdot U^{†}\left(t_{2}-t_{1}\right)\cdot M_{65}^{†}\right]$. Now we turn to the more general open system process that contains both quantum and Markov components.

### 2.3. The Open System Model

An open system model operates on a density matrix operating within the vector space rather than a state vector. The open system model describes the evolution of the density matrix using the following master equation:(1)$$\frac{d}{dt}\rho \left(t\right)=-i\cdot \left(1-w\right)\cdot \left[H,\rho \left(t\right)\right]+w\cdot \sum_{i,j}\gamma_{ij}\cdot \left(\left(L_{ij}\cdot \rho \left(t\right)\cdot L_{ij}^{†}\right)-.5\cdot \left\{\left(L_{ij}^{†}\cdot L_{ij}\right),\rho \left(t\right)\right\}\right).$$

The master equation is a weighted sum of two components: The first component represents the contribution from the quantum evolution and the second component contains what are called the Lindblad operators that generate the Markov contribution. The weight 0≤w≤1 determines the relative importance of each contribution. The coefficients $\gamma_{ij}$ form a matrix *G*, which is required to be positive semi definite to guarantee that the master equation generates a density matrix. The matrices $L_{ij}$ are the Lindblad operators that are discussed below in Section 2.4, and $\left\{\left(L^{†}\cdot L\right),\rho \right\}=\left(L^{†}\cdot L\right)\cdot \rho +\rho \cdot \left(L^{†}\cdot L\right)$. The trace of $\frac{d}{dt}\rho \left(t\right)$ must equal zero so that the trace of the density ρ(t) continues to equal to one across time. This implies that when w=1, the trace of the Lindblad component must be zero.

There are at least two different ways to solve Equation (Equation 1). One way is to directly solve the differential equation, perhaps using numerical methods. A second way, described by [11,22], is to vectorize the state $\varrho =Vec\left[\rho \right]$ by stacking each column of ρ on top of each other to form a $N^{2}\times 1$ vector. (Note that Vec is a linear operation.) Equation (Equation 1) is linear with respect to ρ, which implies that we can rewrite Equation (Equation 1) as a linear differential equation in the form $\frac{d}{dt}\varrho =L\cdot \varrho$, with: $$L\cdot \varrho =-i\cdot \left(1-w\right)\cdot Vec\left([H,\rho ]\right)+w\cdot \sum \gamma_{ij}\cdot \left(Vec\left(L_{ij}\cdot \rho \cdot L_{ij}^{†}\right)-.5\cdot Vec\left\{\left(L_{ij}^{†}\cdot L_{ij}\right),\rho \right\}\right)$$
which has the solution $\varrho \left(t\right)=e^{t\cdot L}\cdot \varrho \left(0\right)$. To identify the operator L, the following tensor identity is useful (see [23], p. 333): $Vec\left[XYZ\right]=\left(Z^{T}⊗X\right)\cdot Vec\left(Y\right)$ where X,Y,Z are matrices, and $Z^{T}$ is the matrix transpose (without conjugation). Then we can write L·ϱ using the following identities:$$\begin{array}{cc} Vec\left(H\cdot \rho \cdot I-I\cdot \rho \cdot H\right) & =\left(I⊗H-H^{T}⊗I\right)\cdot \varrho \\ Vec\left(L_{ij}\cdot \rho \cdot L_{ij}^{†}\right) & =\left(L_{ij}^{*}⊗L_{ij}\right)\cdot \varrho \\ Vec\left(\left(L_{ij}^{†}\cdot L_{ij}\right)\cdot \rho \cdot I\right) & =\left(I⊗\left(L_{ij}^{†}\cdot L_{ij}\right)\right)\cdot \varrho \\ Vec\left(I\cdot \rho \cdot \left(L_{ij}^{†}\cdot L_{ij}\right)\right) & =\left({\left(L_{ij}^{†}\cdot L_{ij}\right)}^{T}⊗I\right)\cdot \varrho \end{array}$$

Collecting these terms together produces:$$\begin{array}{cc} L & =-i\cdot \left(1-w\right)\cdot \left(I⊗H-H^{T}⊗I\right)\\ + & w\cdot \sum \gamma_{ij}\left(\left(L_{ij}^{*}⊗L_{ij}\right)\right)-.5\cdot \left(I⊗\left(L_{ij}^{†}\cdot L_{ij}\right)+{\left(L_{ij}^{†}\cdot L_{ij}\right)}^{T}⊗I\right). \end{array}$$

### 2.4. Application to Cognitive Science

The first main challenge that a cognitive scientist must face when trying to apply this open system model is to define the Lindblad operators $L_{ij}$. We recommend following [11,22], and define $L_{ij}=|i\rangle \langle j|$, where $|i\rangle$ is a column vector with zeros everywhere except for a one located in the row corresponding to the basis vector $|i\rangle$, and $\langle j|$ is a row vector with zeros everywhere except for a one located in the column corresponding to $\langle j|$. Then the operator $L_{ij}$ represents the transition to $|i\rangle$ from $|j\rangle$.

The second main challenge is to select the coefficients $\gamma_{ij}$ that form the matrix *G*. Using $L_{ij}=|i\rangle \langle j|$, these coefficients can be set equal to the transition probabilities $T_{ij}\left(\tau \right)$ of a Markov chain model. This method provides a direct connection to Markov models familiar to cognitive scientists. The coefficients $\gamma_{ij}$ can be set equal to the transition probabilities $T_{ij}\left(\tau \right)$ of a Markov chain model.

Obviously, if we set w=0, then we obtain exactly the original quantum dynamics for the density matrix. To see how the second (Lindblad) component of Equation (Equation 1) is related to a Markov process, we assume w=1.

Using Equation (Equation 1) with w=1, first we examine the contributions to the $\rho_{k,k}$ diagonal element of the density matrix ρ (the following analysis was provided by [11]):$$\begin{array}{cc} \langle k|\sum_{i,j}\gamma_{ij}\cdot L_{ij}\cdot \rho \cdot L_{ij}^{†}|k\rangle & =\langle k|\sum_{i,j}\gamma_{ij}\cdot |i\rangle \langle j|\cdot \rho \cdot |j\rangle \langle i||k\rangle \\ \; & =\sum_{j}\rho_{jj}\cdot \sum_{i}\gamma_{ij}\cdot \langle k|i\rangle \langle i|k\rangle \\ \; & =\sum_{j}\rho_{jj}\cdot \gamma_{kj}, \end{array}$$
$$\begin{array}{cc} \langle k|\sum_{i,j}\gamma_{ij}\cdot L_{ij}^{†}\cdot L_{ij}\cdot \rho |k\rangle & =\langle k|\sum_{i,j}\gamma_{ij}\cdot |j\rangle \langle i|i\rangle \langle j|\rho |k\rangle \\ \; & =\sum_{i}\gamma_{ik}\cdot \rho_{jk}, \end{array}$$
$$\begin{array}{cc} \langle k|\sum_{i,j}\gamma_{ij}\cdot \rho \cdot L_{ij}^{†}\cdot L_{ij}|k\rangle & =\sum_{i,j}\gamma_{ij}\cdot \langle k|\rho |j\rangle \langle i|i\rangle \langle j|k\rangle \\ \; & =\sum_{i}\gamma_{ik}\cdot \rho_{kk}. \end{array}$$

Therefore we obtain the final result:$$\frac{d}{dt}\rho_{kk}\left(t\right)=\sum_{j}\rho_{jj}\cdot \gamma_{kj}-\sum_{i}\gamma_{ik}\cdot \rho_{kk}.$$

If we set G=T(τ), then $\sum_{i}\gamma_{ik}=1$, because the columns of the transition matrix must sum to one. Assuming G=T(τ), if we define ϕ(t) as the diagonal of ρ then:$$\begin{array}{cc} \frac{d}{dt}\phi \left(t\right) & =T(\tau )\cdot \phi (t)-I\cdot \phi (t)\\ \; & =\left(T(\tau )-I\right)\cdot \phi \left(t\right). \end{array}$$

Note that if G=T(τ) then as required $1^{T}\cdot \left(T\left(t\right)-I\right)\cdot \phi \left(t\right)=1^{T}\cdot \left(T(t)\cdot \phi (t)\right)-1^{T}\cdot \phi \left(t\right)=1-1=0$ (where $1^{T}$ is a row vector containing all one’s).

Recall that the continuous time Markov process is based on a generator $K=lim_{\tau \to 0}\frac{T(\tau )-I}{\tau }$, and obeys the equation $\frac{d}{dt}\phi \left(t\right)=K\cdot \phi \left(t\right)$. Comparing this to the final form of the Lindblad equation, $\frac{d}{dt}\phi \left(t\right)=\left(T\left(\tau \right)-I\right)\cdot \phi \left(t\right)$, we see that they are not quite the same.

If instead we set G=K, then $\sum_{i}\gamma_{ik}=0$, and the Lindblad component becomes identical to the Markov process on the diagonal elements of ρ. When w=1, this is not a problem because the diagonals exactly follow the Markov process. But if 0≤w≤1, then this could become a problem because *K* is a negative definite (the columns of *K* sum to zero) and the open system involving both the quantum and Lindblad components are no longer guaranteed to maintain a density matrix across time.

One possible bridge between the two is obtained by setting $G=\left(\frac{T(\tau )}{\tau }\right)$ for a very small value of 0≤τ≤1. Using this assignment, the Lindblad component produces: $\frac{d}{dt}\phi \left(t\right)=\left(\frac{T(\tau )-I}{\tau }\right)\cdot \phi \left(t\right)$. This could be used to approximate *K* but at the same time maintain a positive semi-definite *G*. However, this proposal runs into trouble when we examine the off diagonals.

Returning to Equation (Equation 1) with w=1, next we examine the contributions to the $\rho_{k,l}$ off diagonal element of the density matrix ρ (the following analysis was provided by the second author):$$\begin{array}{cc} \langle k|\sum_{i,j}\gamma_{ij}\cdot L_{ij}\cdot \rho \cdot L_{ij}^{†}|l\rangle & =\langle k|\sum_{i,j}\gamma_{ij}\cdot |i\rangle \langle j|\cdot \rho \cdot |j\rangle \langle i||l\rangle \\ \; & =\sum_{j}\rho_{jj}\cdot \sum_{i}\gamma_{ij}\cdot \langle k|i\rangle \langle i|l\rangle =0 \end{array}$$
$$\begin{array}{cc} \langle k|\sum_{i,j}\gamma_{ij}\cdot L_{ij}^{†}\cdot L_{ij}\cdot \rho |l\rangle & =\langle k|\sum_{i,j}\gamma_{ij}\cdot |j\rangle \langle i|i\rangle \langle j|\rho |l\rangle \\ \; & =\sum_{i}\gamma_{ik}\cdot \rho_{kl} \end{array}$$
$$\begin{array}{cc} \langle k|\sum_{i,j}\gamma_{ij}\cdot \rho \cdot L_{ij}^{†}\cdot L_{ij}|l\rangle & =\sum_{i,j}\gamma_{ij}\cdot \langle k|\rho |j\rangle \langle i|i\rangle \langle j|l\rangle \\ \; & =\sum_{i}\gamma_{ik}\cdot \rho_{kl}. \end{array}$$

Therefore we obtain the final result:$$\frac{d}{dt}\rho_{kl}\left(t\right)=-\rho_{kl}\cdot \left(\sum_{i}\gamma_{ik}\right).$$

If we set G=T(τ), then $\sum_{i}\gamma_{ik}=1$ and $\frac{d}{dt}\rho_{kl}\left(t\right)=-\rho_{kl}$ producing exponential decay of the off diagonals. Alternatively, if we set G=K, then $\sum_{i}\gamma_{ik}=0$ and $\frac{d}{dt}\rho_{kl}\left(t\right)=0$ with no decay of the off diagonals. Finally, if we set $G=\left(\frac{T(\tau )}{\tau }\right)$ then $\frac{d}{dt}\rho_{kl}\left(t\right)=-\frac{\rho_{kl}}{\tau }$, which very rapidly reduces the off diagonals when τ is very small.

A different way to compare models is to examine the probability distributions over time produced by a Markov process versus the Lindblad process. For small τ, the Markov process can be approximated by the equation $\frac{d}{dt}\phi \left(t\right)=\left(\frac{T(\tau )-I}{\tau }\right)\cdot \phi \left(t\right)$ with solution $\phi \left(t\right)=e^{\left(T_{\tau }-I\right)\cdot \left(t/\tau \right)}\cdot \phi \left(0\right)$. The Lindblad component obeys the equation $\frac{d}{dt}\phi \left(t\right)=\left(T\left(\tau \right)-I\right)\cdot \phi \left(t\right)$ with solution $\phi \left(t\right)=e^{(T_{\tau }-I)\cdot t}\cdot \phi \left(0\right)$. This comparison shows that both models produce the same probability distributions, but over different time scales: $\frac{t}{\tau }$ for the Markov process and a slower time *t* for the Lindblad component.

## 3. Examples

A couple of examples are presented to illustrate the predictions from the open system model, Equation (Equation 1), with $L_{ij}=|i\rangle \langle j|$.

Consider a simple N=2 dimensional open system with two possible responses: An up (e.g., no) state represented by $\rho_{11}$ and a down (e.g., yes) state represented by $\rho_{22}$. Suppose w=0.5, $H=\left[\begin{array}{cc} 0 & 1\\ 1 & 1 \end{array}\right]$ and $K=\left[\begin{array}{cc} -\beta & \alpha \\ \beta & -\alpha \end{array}\right]$ and we set G=K. Then Equation (Equation 1) reduces to: $$\begin{array}{cc} 2\frac{d}{dt}\rho & =-i\cdot \left(\left[\begin{array}{cc} 0 & 1\\ 1 & 1 \end{array}\right]\cdot \left[\begin{array}{cc} \rho_{11} & \rho_{12}\\ \rho_{21} & \rho_{22} \end{array}\right]-\left[\begin{array}{cc} \rho_{11} & \rho_{12}\\ \rho_{21} & \rho_{22} \end{array}\right]\cdot \left[\begin{array}{cc} 0 & 1\\ 1 & 1 \end{array}\right]\right)+\left[\begin{array}{cc} \left(\alpha \cdot \rho_{22}-\beta \cdot \rho_{11}\right) & 0\\ 0 & -\left(\alpha \cdot \rho_{22}-\beta \cdot \rho_{11}\right) \end{array}\right]\\ = & \left[\begin{array}{cc} -i\cdot \left(\rho_{21}-\rho_{12}\right)-\rho_{11}\cdot \left(\alpha +\beta \right)+\alpha & -i\cdot \left(1-2\rho_{11}-\rho_{12}\right)\\ -i\cdot \left(2\rho_{11}-1+\rho_{21}\right) & -i\cdot \left(\rho_{12}-\rho_{21}\right)+\rho_{11}\left(\alpha +\beta \right)-\alpha \end{array}\right] \end{array}$$

This model starts out oscillating like a quantum process, but eventually converges to the equilibrium of a Markov process. Figure 1 shows the probability of the down state (e.g., yes response) as a function of time. The black curve represents the probabilities generated by the 2-dimensional process with G=K. The equilibrium state is obtained as follows. First, $2\frac{d}{dt}\rho_{21}=0$ implies that $\left(2\rho_{11}-1\right)+\rho_{21}=0,$ which implies $\rho_{21}=1-2\rho_{11},$ and also (because ρ is Hermitian) that $\rho_{21}-\rho_{12}=0.$ Next $2\frac{d}{dt}\rho_{22}=0$ implies $\rho_{11}\cdot \left(\alpha +\beta \right)-\alpha =0$ so that $\rho_{11}=\frac{\alpha }{\alpha +\beta }$ and $\rho_{22}=\frac{\beta }{\alpha +\beta }$, which exactly matches the asymptotic result obtained when w=1, which produces a pure Markov process (see Figure 1, red curve). Note also that the asymptotic off diagonal element $\rho_{21}=1-2\cdot \rho_{1,1}=\rho_{22}-\rho_{11}=\frac{\beta -\alpha }{\alpha +\beta }$. Thus the system converges to a coherent density matrix. Without the Lindblad contribution, convergence to an equilibrium state, independent of the initial state, would not be possible. If $\rho_{21}=1-2\rho_{11}$ in the density matrix, then the density is generated by an eigen state of *H*, and the initial and final states remain the same.

Alternatively, suppose w=0.5, $H=\left[\begin{array}{cc} 0 & 1\\ 1 & 1 \end{array}\right]$ and $T=e^{K\cdot \tau }$, with τ sufficiently large to reach the equilibrium transition matrix $T=\frac{1}{\alpha +\beta }\left[\begin{array}{cc} \alpha & \alpha \\ \beta & \beta \end{array}\right]$ and we set G=T. The green curve in Figure 1 shows the probability of the down state (e.g., the yes response) as a function of time for the 2-dimensional process with G=T(τ). Then Equation (Equation 1) reduces to:$$\begin{array}{cc} 2\frac{d}{dt}\rho & =-i\cdot \left(\left[\begin{array}{cc} 0 & 1\\ 1 & 1 \end{array}\right]\cdot \left[\begin{array}{cc} \rho_{11} & \rho_{12}\\ \rho_{21} & \rho_{22} \end{array}\right]-\left[\begin{array}{cc} \rho_{11} & \rho_{12}\\ \rho_{21} & \rho_{22} \end{array}\right]\cdot \left[\begin{array}{cc} 0 & 1\\ 1 & 1 \end{array}\right]\right)+\left[\begin{array}{cc} \frac{\alpha }{\alpha +\beta }-\rho_{11} & -\rho_{12}\\ -\rho_{21} & \frac{\beta }{\alpha +\beta }-\rho_{22} \end{array}\right]\\ \; & =\left[\begin{array}{cc} -i\cdot \left(\rho_{21}-\rho_{12}\right)+\frac{\alpha }{\alpha +\beta }-\rho_{11} & i\cdot \left(1-2\rho_{11}-\rho_{12}\right)-\rho_{12}\\ -i\cdot \left(2\rho_{11}-1+\rho_{21}\right)-\rho_{21} & -i\cdot \left(\rho_{12}-\rho_{21}\right)+\frac{\beta }{\alpha +\beta }-\rho_{22} \end{array}\right]. \end{array}$$

In this case, the equilibrium state is obtained as follows. First, $2\frac{d}{dt}\rho_{21}=0$ implies that $\rho_{21}=-0.5\cdot \left(1+i\right)\cdot \left(2\cdot \rho_{11}-1\right)$. Note that $i\cdot \left(\rho_{21}-\rho_{12}\right)=1-2\cdot \rho_{11}$. Then, $2\frac{d}{dt}\rho_{11}=0$ implies that $\rho_{11}=\frac{1}{3}\left(1+\frac{\alpha }{\alpha +\beta }\right)$, and $\rho_{22}=\frac{1}{3}\left(1+\frac{\beta }{\alpha +\beta }\right)$, and the latter falls below the asymptote $\frac{\beta }{\alpha +\beta }$ of a pure Markov process. Finally, $\rho_{21}=\frac{\left(1+i\right)}{6}\cdot \left(1-2\cdot \frac{\alpha }{\alpha +\beta }\right)$.

Now consider another example with a large (N=101) number of levels of evidence. For a pure Markov process, we use a generator that has the following tridiagonal form to produce a Markov random walk:$$K=\left[\begin{array}{ccccc} \;-\beta & \alpha & 0 & & 0\\ \beta & -\lambda & \ddots \\ 0 & \beta & \ddots & \alpha & 0\\ & & \ddots & -\lambda & \alpha \\ 0 & & 0 & \beta & \;-\alpha \end{array}\right].$$

The mean drift rate for this Markov random walk is defined by μ=(β−α)/2 and the diffusion rate is defined by $\sigma^{2}=\left(\beta +\alpha \right)/2$. In this example, we used a positive mean drift rate μ>0 that pushes the probability distribution to the right (high evidence levels). This generator uses reflecting bounds, which produces a unique invariant distribution [20].

For a pure quantum process, we use a Hamiltonian that has the following tridiagonal form to produce a quantum walk:$$H=\left[\begin{array}{ccccc} \mu_{1} & \sigma & 0 & & 0\\ \sigma & \mu_{2} & \ddots \\ 0 & \sigma & \ddots & \sigma & 0\\ & & \ddots & \mu_{N-1} & \sigma \\ 0 & & 0 & \sigma & \mu_{N} \end{array}\right].$$

The diagonal contains the potential function, μ(x), which can be defined by a quadratic potential $\mu \left(x\right)=a+b\cdot x+c\cdot x^{2}$. In this example, we simply used a linearly increasing potential, b>0,a=0,c=0, that pushes the distribution of squared amplitudes toward the right (high levels of evidence). Once the wave hits the reflecting bound, it bounces back producing oscillation and interference. This pure quantum process never reaches an invariant state [19].

Below we compare four different models (see https://jbusemey.pages.iu.edu/quantum/HilbertSpaceModelPrograms.htm for the Matlab program used to make these computations). A pure Markov model, a pure quantum model, an open system model with G=K (the generator for the pure Markov process), and an open system model with G=T(τ). To match the time scales of the pure Markov process and the open system model with G=T(τ), we set the time scale of the latter to t/τ. The initial distribution, ϕ(0), was a discrete approximation to a Gaussian distribution centered at the middle of the evidence scale for a pure Markov process and the square root of this distribution was used for the initial state ψ(0) of the quantum and open system models.

First we examine the open system model when w=1. This should reduce the two open system models to a pure Markov process. (For the open system with G=T(τ), we set $\tau =10^{-3}$ to obtain a good approximation to the pure Markov process.) The left panel of Figure 2, shows the probability distribution over evidence levels produced at a moderately short time interval. Only two curves can be seen. One is the bell-shaped curve produced by the pure quantum model. The potential function moved the distribution of squared amplitudes from the initial state in the middle, 0.50, up toward the right with a mode around 0.70. The other curve actually includes three overlapping curves produced by the pure Markov model and the two open system models with w=1. This simply shows that the open system does indeed reproduce the Markov model when the quantum component is eliminated.

Next we examine the open system model when w=0.5. The right panel of Figure 2 shows the mean evidence across time produced by the pure quantum model, the pure Markov model, the open system model with G=K, and the open system model with G=T(1). As can be seen in the right panel, the pure quantum process oscillates between 0.5 and 0.75 indefinitely. The pure Markov process monotonically increases to an asymptote equal to 0.86. The open system model with G=K starts out oscillating like the quantum model, but then converges to the same equilibrium, 0.86, of the Markov model. The the open system model with G=T(1) starts out oscillating like the quantum model, but then converges to a lower equilibrium 0.82 than the Markov model. The supplement provides the first order equations that need to be satisfied for equilibrium (see Appendix A).

## 4. Summary and Concluding Comments

For over 50 years cognitive scientists have successfully modeled choice probability and distribution of choice times for human decision making using Markov random walk or diffusion processes. However, there is new empirical evidence that these models may not be sufficient and quantum walks may be needed to capture some behavior that can not easily be explained by Markov processes. This new evidence includes interference effects of choice on later confidence and temporal oscillations in preference. Thus both types of processes may be needed and a promising way to combine the two processes is by using open system models. An open system combines quantum and Markov components into a single dynamic process.

One might argue that a simpler way to combine Markov and quantum processes together is simply to form a weighted average of the two separate probability distributions produced by the two separate processes. This is quite different from an open system, which computes a single probability distribution from a single unified process containing both quantum and Markov components. We think the open system is preferred for two important reasons. One is that open systems provide the proper dynamics by starting out in a quantum oscillatory regime and later converge to a Markov regime that reaches an equilibrium. Simply averaging the two processes would create dynamics in which both processes are always operating and present. In particular, a weighted average would continue oscillating indefinitely and never converge to an equilibrium. A second reason is the interpretation of the systems. An open system describes the dynamics of two different types of uncertainty: Epistemic uncertainty in which an outside observer is uncertain about the state of the decision maker, and ontic uncertainty in which the decision maker is in a superposition state over evidence. A simple weighted average would imply that a person’s state is somehow switching from time to time from an epistemic type of uncertainty about the state to an ontic uncertainty about the state.

In this article we reviewed pure Markov random walks, quantum walks, and open systems. We also reviewed two different methods for computing the predictions of an open system: One is to numerically solve the differential equations, and the second is to vectorize the matrix system. We recommend that latter because it provides predictions that can be computed directly from a matrix exponential function.

We also covered important challenges toward applications of open systems to cognitive science. One is the choice of Lindblad operators that form the Lindblad or Markov component of the open system. We recommend following the suggestion by [11] to use the basic transition operators $L_{ij}=|i\rangle \langle j|$ which describe the transitions to state *i* from state *j*, making the model similar to Markov chains that are familiar to cognitive scientists.

A second challenge is to define the Lindblad coefficients that form the matrix *G*. This turned out to be a bit more complicated. On the one hand, one could set G=K, where *K* is the generator of a continuous time Markov process. This has the advantage of reducing as a special case directly to a continuous time Markov process when the full weight is applied to the Lindblad component. The disadvantage is that it is no longer guaranteed to produce a density matrix across time. The trace always sums to unity but the diagonals could go negative. In all the examples that we have examined, this has not been a major problem, but it could happen. On the other hand, one could set G=T(τ), where T(τ) is the transition matrix produce by the generator *K*. This has the advantage of guaranteeing the system always generates a density matrix. However, it has the disadvantage of requiring one to estimate an additional parameter τ, and it does not reduce to a continuous time Markov process when the full weight is applied to the Lindblad operator. Instead it operates on a time scale inversely related to τ. It is too early to say which choice of *G* is best. We recommend trying both ways at this stage. That way one can try G=K and check to see if this causes problems with the density and also try G=T(τ) and check to see if the time scale becomes a problem. Furthermore, because the predictions from the two choices for *G* will not be the same, one can check to see which one actually accounts for the behavioral data best.

Finally one advantage of using the open system as opposed to only the Markov system or only the quantum system is that the fit of the model to data can be used to determine the weight *w* on each component. If the quantum system is not needed, this weight will reduce to one. However, in applications so far, substantial weight (w≈0.7 on the Lindblad) has been needed to account for the findings [7,11].

## Figures and Tables

**Figure 1 entropy-22-00990-f001:**
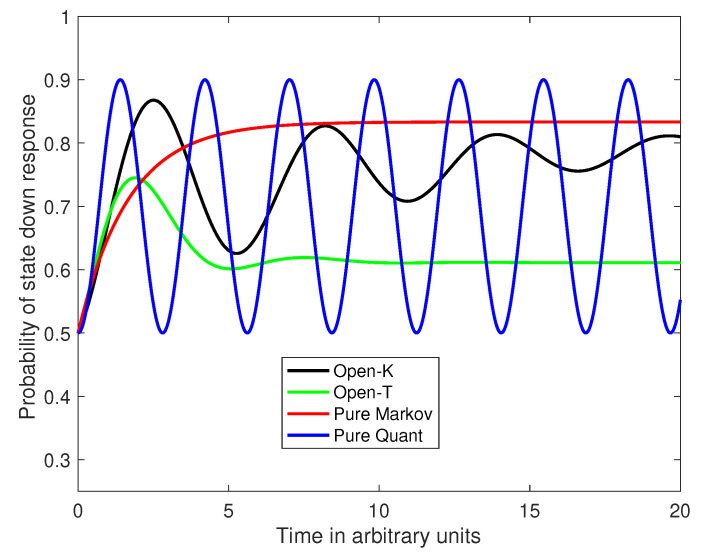
Probability of responding ‘yes’ as a function of time for the 2 dimensional Markov, quantum, open systems using G=K, and G=T.

**Figure 2 entropy-22-00990-f002:**
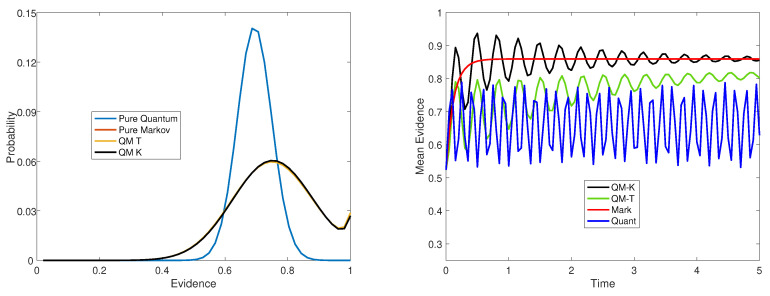
Left panel: Probability distribution across levels of evidence when w=1 for the open system. Right panel: Mean evidence across time when w=0.5 for the open system.

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
