# Peer review of "Application of Quantum—Markov Open System Models to Human Cognition and Decision"

_entropy, 2020, doi:10.3390/e22090990_

Round 1
Reviewer 1 Report
In my opinion, the work is important and well written. However, I see a drawback. This article deals with the decision problem, but the authors do not mention quantum game theory. I think it is important to characterize other methods in the introduction. Game theory has been linked to decision theory from the beginning. See eg. "Quantum Games and Quantum Strategies". Physical Review Letters. 83 (15). There are interesting results about preferences (transitive or intransitive), eg. „Transitivity of an entangled choice”. J. Phys. A Math. Theor. 2011, „Do transitive preferences always result in indifferent divisions?”, Entropy 2015 or When ‘I cut, you choose’ method implies intransitivity. Phys. A 350C, 189–193 (2014). There are also interesting results regarding evolutionary processes eg „Evolutionary Processes in Quantum Decision Theory”, Entropy 2020, 22(6).
I think it is important that the reader also has a shortened view of the other results (necessarily only brief)
Author Response
We added these references
Do transitive preferences always result in indifferent divisions?”, Entropy 2015 or When ‘I cut, you choose’ method implies intransitivity. Phys. A 350C, 189–193 (2014). There are also interesting results regarding evolutionary processes eg „Evolutionary Processes in Quantum Decision Theory”, Entropy 2020, 22(6).
However, game theory is outside the domain of the topic of this article
Reviewer 2 Report
I am happy to recommend this article for publication in Entropy. It is an effective and important article, marked by conceptual and technical sophistication of its argument. Its main contribution, a consideration of an open system that combines Markov and quantum component is an important innovation in the field of cognitive decision making.
I do, however, have one concern. I think that in the following defining statement "Quantum models represent an ontic type of uncertainty in which the decision maker is in a superposed state with internal uncertainty about his or her location of evidence" repeated three times (ll. 777-78, 160-161, 293-294), the concept of a superposed state of the decision maker and internal uncertainty about his or her location of evidence need to be explained in more detail. It will, in my view, not be self-evident as stated for many readers. A more informed reader could surmise this, for example, from the discussion of representation of states by density operators in 2.2, but a brief general explanation and an example would be helpful in my view.
Author Response
we elaborated on the ontic nature of the quantum superposition
Quantum models represent an ontic type of uncertainty in which there is no preexisting location of evidence before a decision is made. Instead, at each moment, the decision maker is in a superposed state with several different levels of evidence having potential to be realized, so that the decision maker has internal uncertainty about the level of evidence.
Reviewer 3 Report
The aim of the authors is to address the challenges in modelling the dynamics of human choice and confidence across time. For this purpose they recommend to resort to the open-system models. They review pure Markov random walks, quantum walks, and open systems. They also review two different methods for computing the predictions of an open system.One way is to numerically solve the differential equations, and the second is to vectorize
the matrix system. Some important challenges toward applications of open systems to cognitive science are discussed.
One comment: in lines 284 and 285, it seems, there is a misprint, where we see: "to form a weighted average of the two separate probability distribution produced by the two separate process". It should be here: "distributions" and "processes".
The article suggests a useful survey of the open-system approach to modelling the dynamics of cognitive processes. It can be accepted for publication in its present form.
Author Response
Thanks we made this change
It should be here: "distributions" and "processes".
we also added a reference to another related article in this special issue
Yukalov, V.I. Evolutionary Processes in Quantum Decision Theory. Entropy 2020, 22.
353 doi:10.3390/e22060681